# Underreporting and Triggering Factors for Reporting ADRs of Two Ophthalmic Drugs: A Comparison between Spontaneous Reports and Active Pharmacovigilance Databases

**DOI:** 10.3390/healthcare10112182

**Published:** 2022-10-31

**Authors:** Homero Contreras-Salinas, Leopoldo Martín Baiza-Durán, Manuel Alejandro Bautista-Castro, Diego Rodrigo Alonso-Rodríguez, Lourdes Yolotzin Rodríguez-Herrera

**Affiliations:** Pharmacovigilance Department, Laboratorios Sophia, S.A. de C.V., Zapopan 45010, Jalisco, Mexico

**Keywords:** adverse drug reaction, spontaneous reporting, active surveillance, underreporting, antiglaucoma, artificial tears

## Abstract

(1) Aims of the study: calculating the underreporting ratio for two different medications, a fixed combination of 0.5% timolol + 0.2% brimonidine + 2.0% dorzolamide (antiglaucoma) and a fixed combination of sodium hyaluronate 0.1% + chondroitin sulfate 0.18% (artificial tears) for characterizing the features influencing the reporting of adverse drug reactions (ADRs) in spontaneous reporting. (2) Methods: The underreporting ratio was calculated by comparing the adverse drug reactions reported in the spontaneous reporting database for every 10,000 defined daily doses marketed and the adverse drug reactions from an active surveillance study for every 10,000 defined daily doses used for different drugs (antiglaucoma and artificial tears). The factors related to the report in spontaneous reporting through statistical tests were also determined. (3) Results: The underreporting ratio of spontaneous reporting was 0.006029% for antiglaucoma and 0.003552% for artificial tears. Additionally, statistically significant differences were found for severity, unexpected adverse drug reactions, and incidence of adverse drug reactions in females when compared with spontaneous reporting and active surveillance. (4) Conclusions: The underreporting ratio of ADRs related to ophthalmic medications indicates worry since the cornerstone of pharmacovigilance focuses on spontaneous reporting. Additionally, since underreporting seems to b selective, the role of certain aspects, such as gender, seriousness, severity, and unexpected ADRs, must be considered in future research.

## 1. Introduction

Identifying adverse drug reactions (ADRs) is essential for pharmacovigilance activities; it determines the actions to be followed in order to prevent risks [1]. ADRs are identified during early preclinical stages and in randomized controlled trials; however, it is challenging to detect rare ADRs; thus, post-marketing studies are crucial [2,3]. These considerations pose a challenge in detecting rare ADRs and drug interactions; therefore, not all risks are identified at the time of initial marketing authorization [1,4], and the implementation of strategies for the collection of post-marketing ADRs is consequently essential [2,5,6].

Spontaneous reporting is the most common form of post-marketing pharmacovigilance. It is characterized by no active actions to scrutinize ADRs apart from the encouragement of health professionals and patients to report them; therefore, such reports depend entirely on their initiative and motivation [7]. Spontaneous reporting has some limitations, such as underreporting, variable quality of the reported data, and lack of information on drug exposure [4,8].

One of the main reasons for underreporting in spontaneous reporting is that patients and health professionals do not believe it necessary to register an already known or well-established ADR [9]. In addition, the absence of time, different care priorities, uncertainty about the drug causing the ADR, difficulty in accessing notification formats, and a lack of understanding of the purpose of notification can also account for ADR underreporting [10]. Consequently, with spontaneous reporting, it is not likely to identify the true frequency of an ADR, nor does it allow for an optimal appreciation of ADRs in new patients and/or populations [9].

Underreporting is a concern worldwide because it causes the frequency of ADRs to be underestimated, delaying the detection of safety issues and therefore making it difficult to undertake timely action [5,11]. Nevertheless, unlike spontaneous reporting, other tools, such as active surveillance, intentionally, and continually search for the presence or absence of ADRs in a defined group of people; it thus allows the real incidence of ADRs to be known. Moreover, active pharmacovigilance provides data on the risk factors associated with medication and allows for more efficient use of such products [12].

This research aimed to calculate the underreporting ratio in two different medications, a fixed combination of 0.5% timolol + 0.2% brimonidine + 2.0% dorzolamide (antiglaucoma) and a fixed combination of sodium hyaluronate 0.1% + chondroitin sulfate 0.18% (artificial tears), and to characterize the features that influenced the reporting of ADRs in spontaneous reporting.

## 2. Materials and Methods

### 2.1. Study Design

Two collection ADR reporting methods were compared (spontaneous reporting and active surveillance) by analyzing the data of two marketed products.

### 2.2. Spontaneous Reporting

The data for spontaneous reporting were collected from the ADRs referred to the Pharmacovigilance and Technovigilance Unit of Laboratorios Sophia S.A. de C.V over Jan 2017 to July 2022 in countries where artificial tears (México, Colombia, Bolivia, Uruguay, Costa Rica, Ecuador, El Salvador, Guatemala, Haití, Honduras, Nicaragua, Panamá, Perú and Venezuela) and antiglaucoma (México, El Salvador, Costa Rica, Guatemala, Nicaragua, Colombia, Panamá, Haití, Chile, República Dominicana, Bolivia, Ecuador and Honduras) are commercialized. Only spontaneous reports were included.

### 2.3. Active Surveillance Reporting

The data were gathered from two non-intervention active surveillance programs in Peru (antiglaucoma and artificial tears [all patients prescribed under the doctor’s own decision, with signed informed consent joined the study]) and safety information collected through three follow-up calls after 5, 30 and 60 days, respectively. During the initial contact call, the patients were questioned about their personal data, drug prescription characteristics, clinical history, and identified ADRs. The second and third contacts were aimed exclusively at identifying ADRs.

### 2.4. Categorization

In both databases, the patients were classified as children (0–12 years old [yo]), adolescents (>12–18 yo), adults (>18–60 yo), or elders (>60 yo). ADRs were listed according to the Medical Dictionary for Regulatory Activities (MedDRA) in System Organ Class (SOC) and Preferred Term (PT), while causalities were accomplished by the WHO-UMC method [13]. Severity was evaluated using the ADR Severity Assessment Scale (Hartwig and Siegel modified) in Mild, Moderate, or Severe [14,15], and seriousness was assessed by the ADR’s nature to generate permanent damage or to be considered life-threatening [16].

### 2.5. Calculating Defined Daily Dose Indicators

The “Utilization in Defined Daily Dose (DDD)” for spontaneous reporting was calculated considering the marketed pieces of the product (January 2017 to July 2022) using the World Health Organization formula [17].
Utilization in DDD = (Number of packages)(Number of DDDs in a package)

The “utilization in DDD” from the active surveillance study was calculated considering one DDD/day per patient during the study period (30 or 60 days), and this was multiplied by the number of patients admitted.
Utilization in DDD = (Days using DDDs)(Number of patients in active surveillance program)

Finally, “ADRs per 10,000 DDDs” was calculated to visualize how many ADRs arose for every 10,000 DDDs.
(1)$$ADRs\;per\;10,000\;DDDs=\frac{\mathrm{ADRs}}{\mathrm{Utilization}\;\mathrm{in}\;\mathrm{DDD}}\;\left(10,000\right)$$

Underreporting ratio

The underreporting ratio was calculated using the formula mentioned below.
(2)$$Underreporting\;ratio=\frac{ADRs\;per\;10,000\;DDDs\;\left(spontaneous\;reporting\right)}{ADRs\;per\;10,000\;DDDs\;\left(active\;surveillance\right)}$$

### 2.6. Factors Related to Reporting

Age group, gender, severity, seriousness, causality, and the different ADRs identified from both databases (spontaneous and active surveillance) and whether the ADRs appeared in the summary of each product’s characteristics (SmPC)/monograph (label/unlabeled), were analyzed to determine which factors were involved in reporting rate.

### 2.7. Statistical Analysis

Qualitative variables were described as frequencies and percentages. Chi-square or Fisher’s exact tests (small samples) were performed for the inter-group analysis. This study’s statistical significance was 2-sided set at a *p*-value ≤ 0.05. SPSS (version 21; SPSS, Inc., Chicago, IL, USA) was used for the statistical analysis.

## 3. Results

Spontaneous reporting from January 2017 to July 2022 was distributed across 195,334,179 DDDs for the antiglaucoma (75 ADRs notified) and 133,029,769 for the artificial tears (26 ADRs notified), representing 0.003840 and 0.001955 ADRs for every 10,000 DDDs sold, respectively (Table 1). However, during the 60-day antiglaucoma (246 patients and 94 ADRs notified) and 30-day artificial tears (212 patients and 35 ADRs notified) active surveillance studies, 63.69 and 55.03 ADRs per every 10,000 DDD were reported, respectively (Table 2). The underreporting ratio of spontaneous reporting was calculated with the data described above, resulting in 0.006029% for antiglaucoma and 0.003552% for artificial tears. According to these calculations, artificial tears have a nearly twofold underreporting ratio compared to antiglaucoma.

The age groups included in the active surveillance study for antiglaucoma were children (n = 1; 0.41%), adults (n = 96; 39.02%), and elders (n = 149; 60.57%). Meanwhile, for the artificial tears study, they were children (n = 1; 0.47%), adolescents (n = 2; 0.94%), adults (n = 110; 51.89%), and elders (n = 99; 46.70%). As expected, antiglaucoma was mostly prescribed to elderly patients, whereas artificial tears, mainly in the adult group, were exposed. The information collected from active surveillance and spontaneous reporting showed that, for the antiglaucoma study, the age group presenting ADRs most frequently was the elders. In contrast, the age group with the most frequent ADR presentation for the artificial tears group were adults, congruent with the age groups where each of the medications were mostly prescribed (Figure 1).

A statistically significant increase in severity was found for both spontaneous reporting databases (antiglaucoma and artificial tears) when they were contrasted with the active surveillance database [(X^2^_(2)_ = 27.57, *p* < 0.0001) antiglaucoma], [(Fisher´s exact test, *p* = 0.0063) artificial tears]. A non-statistically significant trend was observed for the increase in ADR seriousness in spontaneous reporting compared to active surveillance for the antiglaucoma database (Fisher´s exact test, *p* = 0.1955). No serious adverse reactions were identified in the artificial tears study database (Table 3).

An increased number of ADRs was found in females compared to males in the active surveillance database: 19.2% in the antiglaucoma group and 14.2% more in the artificial tears group. In addition, when contrasting these results with the spontaneous reporting database, the increase in the presence of ADRs in females was superior to that of active surveillance, with a rise of 1.1-fold in the antiglaucoma group and 1.9-fold more in the artificial tears group (Table 3).

A total of 11 PTs were identified in the databases (active surveillance) compared to 36 PTs (spontaneous reporting) for the antiglaucoma group (Table 4), and 4 PT (active surveillance) compared to 13 PTs (spontaneous reporting) for the artificial tears group (Table 5).

In the antiglaucoma database, the most frequent ADR was eye irritation, encompassing 80.9% of all active surveillance ADRs, as opposed to the spontaneous reporting database, with a frequency of 25.3% of the total number of ADRs (Table 4). In the case of artificial tears treatment databases, the outcome was similar to that of antiglaucoma, where the most frequent ADR (burning sensation) showed a significantly higher incidence in the active surveillance database, with 74.3% of the ADRs reported compared to 42.3% of the total ADRs notified in spontaneous reporting (Table 5).

The most frequent causality for active surveillance (both treatments) was “probable”; however, for this parameter, a marked difference was observed in the reports of active surveillance when compared to spontaneous reporting in both drug groups: antiglaucoma (86.17% vs. 16.00%) and artificial tears (71.43% vs. 42.30%) (Figure 2).

In addition, the spontaneous reporting databases showed a statistically significant increase in ADRs not included in the SmPC/monograph (unlabeled). For antiglaucoma’s spontaneous reporting, 56 ADRs were labeled, while 19 were unlabeled. For this same group’s active surveillance, 91 were labeled, and 3 were unlabeled (X^2^_(1)_ = 18.06, *p* < 0.0001) (Table 4). For the artificial tears, spontaneous reporting included 13 labeled, and 13 unlabeled, while active surveillance included 33 labeled, and 2 unlabeled (X^2^_(1)_ = 15.29, *p* < 0.0001) (Table 5).

The lack of information on causality, severity, age group, gender, and seriousness in four reports from artificial tears, as well as causality, severity, and age group in six reports from antiglaucoma in the spontaneous reporting database, is worth mentioning (Table 3 and Figure 2).

## 4. Discussion

The outcomes of the research show significant rates of underreporting (0.006029% for antiglaucoma and 0.003552% for artificial tears); this indicates a worrying result since the pillar of pharmacovigilance focuses on spontaneous reporting [18]; additionally, underreporting limits the obtainment of sufficient data to generate risk minimization activities, representing a significant threat to the population [19]; for this reason, there is a need to encourage reports in pharmacovigilance, especially in ophthalmics due to is evidence that the notifications in ophthalmology are reduced compared to other medical specialties [20,21]. Additionally, underreporting seems to be selective for some factors, such as seriousness, severity, gender, and unexpected ADRs.

As previously mentioned, several authors have described that one of the problems of spontaneous reporting is derived from the fact that patients only notify ADRs when they are severe or serious, leaving aside notifications of less severe and serious events [22,23,24]. This agrees with what is observed when comparing the spontaneous reporting databases concerning active surveillance, where a statistically significant difference was observed in the presence of severe ADRs in the spontaneous reporting database and with evidence of a tendency for seriousness, with a higher number of notifications of serious/severe ADRs. Furthermore, the difference in the underreporting ratio, which was nearly twofold for the artificial tears, could be due to this effect. Since the severity/seriousness of adverse reactions in the artificial tears database is lower than for the antiglaucoma (Table 3).

Similar to what was found in this research, an increase in ADRs in females has already been reported. The literature divides this event into two main aspects: those related to gender (more informed on health issues, concern about your health, a propensity to report symptoms, increased interest in reporting) [25,26], and those related to sex (metabolic, pharmacokinetic, pharmacodynamic, and hormonal factors) [27,28]. It is possible that factors related to sex could influence the increase in ADRs in the active surveillance database. In contrast, factors associated with gender could not influence this type of study because the active surveillance responsibility for collecting ADRs rests on the sponsor (monitoring by the study period). Nevertheless, an increase was observed in spontaneous reporting compared to the active surveillance database. In this case, the factors related to gender are not inconsequential. This could explain the observed increase in the spontaneous reporting database; added to this, the relationship of gender in ophthalmic medication-related ADRs has already been reported in a previous article that found a statistically significant difference between males and females in a drug safety surveillance study in a dexamethasone/ciprofloxacin ophthalmic solution [29].

A statistically significant difference was found between unexpected and expected ADRs in the spontaneous reporting database; this agrees with what has been previously reported in the literature. It has been mentioned that patients tend to notify ADRs only when they are unexpected [22,24].

The low number of ADRs in the spontaneous reporting database for antiglaucoma (“eye irritation” 25.3% incidence in spontaneous reporting vs. 80.9% in active surveillance) and in the artificial tears database (“burning sensation” 42.3% (spontaneous reporting) vs. 74.3% (active surveillance) (Table 4 and Table 5) highlights the low incidence of spontaneously reported ADRs due to lack of seriousness (not serious), low severity (mild), frequency, and expectedness.

Finally, four ADRs for artificial tears and six reports for antiglaucoma lack data in the spontaneous reporting database; this problem has already been documented in the literature [4,8], where spontaneous reports usually lack information due to the impossibility of follow-up.

The study’s main limitations were the low number of ADRs analyzed and the relatively small sample size of active surveillance studies, as well as the use of DDD as a mechanism to determine the rate of underreporting due to the possibility that not all distributed doses were actually administered or that not all patients maintained a posology as described by the DDD formula.

## 5. Conclusions

The underreporting ratio of ADRs related to ophthalmic medications indicates worry since the cornerstone of pharmacovigilance focuses on spontaneous reporting. Besides, since underreporting seems to be selective, the role of certain aspects such as gender, seriousness, severity, and unexpected ADRs must be considered in future research.

## Figures and Tables

**Figure 1 healthcare-10-02182-f001:**
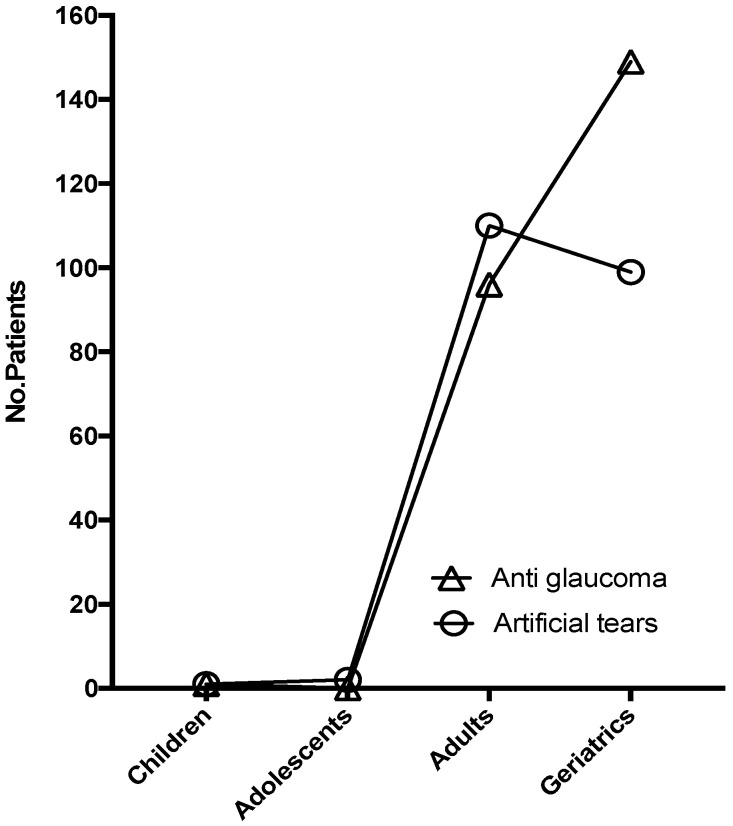
Age group of patients admitted to different active surveillance studies.

**Figure 2 healthcare-10-02182-f002:**
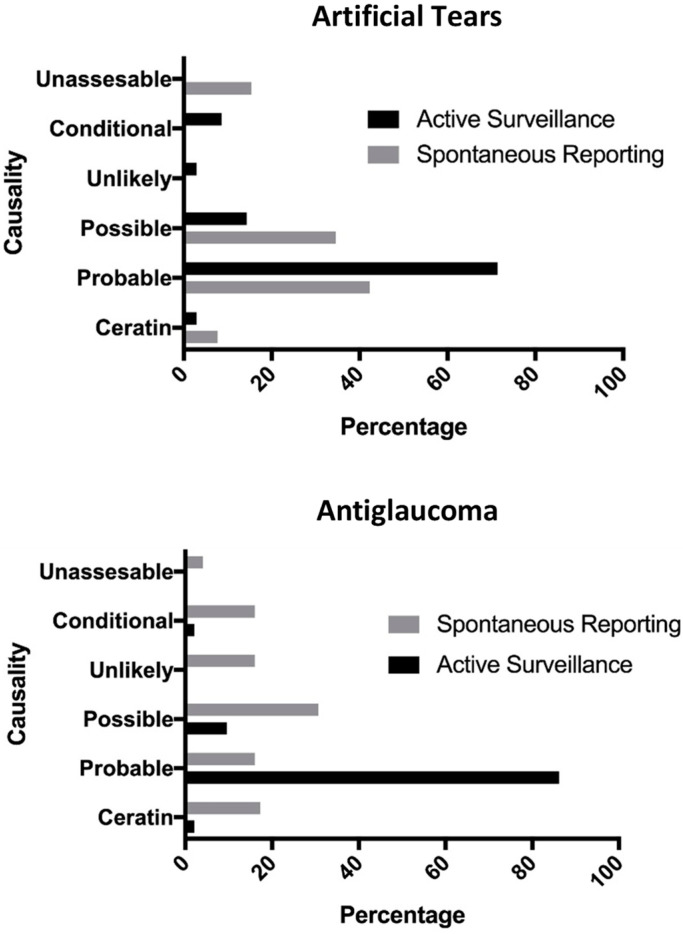
Comparison of causalities between spontaneous reporting and active surveillance in the glaucoma and artificial tears databases.

**Table 1 healthcare-10-02182-t001:** Defined Daily Dose of Spontaneous Notification.

Drug	Time	ADRs	DDD	Use in DDD	ADRs per 10,000 DDDs
Antiglaucoma	2005 days	75	0.14 mL	195,334,179	0.003840
Artificial tears	2005 days	26	0.54 mL	133,029,769	0.001955

ADRs, Adverse Drug Reactions. DDD, Defined Daily Dose.

**Table 2 healthcare-10-02182-t002:** Defined Daily Dose of Active Surveillance.

Drug	Time	Patients	ADRs	DDD	Use in DDD	ADRs per 10,000 DDDs
Antiglaucoma	60 days	246	94	0.14 mL	14,760	63.68563686
Artificial tears	30 days	212	35	0.54 mL	6360	55.03144654

ADRs, Adverse Drug Reactions. DDD, Defined Daily Dose.

**Table 3 healthcare-10-02182-t003:** Characteristics of patients´ ADRs from the databases.

		Antiglaucoma		Artificial Tears	
		AS, n (%)	SN, n (%)	AS, n (%)	SN, n (%)
Age group	Children	0 (0)	1 (1.3)	0 (0)	0 (0)
	Adolescents	0 (0)	0 (0)	0 (0)	0 (0)
	Adults	43 (45.7)	16 (21.3)	22 (62.9)	18 (69.2)
	Geriatrics	51 (54.3)	52 (69.4)	13 (37.1)	4 (15.4)
	Unknown	0 (0)	6 (8)	0 (0)	4 (15.4)
Gender	Male	38 (40.4)	30 (40.0)	15 (42.9)	8 (30.8)
	Female	56 (59.6)	45 (60.0)	20 (57.1)	14 (53.8)
	Unknown	0 (0)	0 (0)	0 (0)	4 (15.4)
Severity	Mild	94 (100)	51 (68.0)	35 (100)	17 (65.4)
	Moderate	0 (0)	16 (21.3)	0 (0)	5 (19.2)
	Severe	0 (0)	2 (2.7)	0 (0)	0 (0)
	Unknown	0 (0)	6 (8)	0 (0)	4 (15.4)
Seriousness	Not-Serious	94 (100)	73 (97.3)	35 (100)	22 (84.6)
	Serious	0 (0)	2 (2.7)	0 (0)	0 (0)
	Unknown	0 (0)	0 (0)	0 (0)	4 (15.4)
Total	ADRs	94	75	35	26

Surveillance. SN, Spontaneous Notification.

**Table 4 healthcare-10-02182-t004:** Most frequently reported ADRs in antiglaucoma databases.

Antiglaucoma			
PT	Labeled *	AS, n (%)	SN, n (%)
Eye irritation	Y	76 (80.9)	19 (25.3)
Vision blurred	Y	5 (5.3)	6 (8.0)
Lacrimation increased	Y	0 (0)	5 (6.7)
Eye pain	Y	1 (1.1)	4 (5.3)
Eye inflammation	Y	0 (0)	4 (5.3)
Headache	N	1 (1.1)	3 (4.0)
Ocular hyperaemia	Y	1 (1.1)	3 (4.0)
Eye pruritus	Y	3 (3.2)	2 (2.7)
Somnolence	Y	1 (1.1)	2 (2.7)
Blood pressure decreased	Y	1 (1.1)	2 (2.7)
Diffuse alopecia	N	0 (0)	2 (2.7)
Eyelid skin dryness	N	0 (0)	2 (2.7)
Cough	N	0 (0)	2 (2.7)
Dysgeusia	Y	2 (2.1)	1 (1.3)
Dry mouth	Y	1 (1.1)	1 (1.3)
Arthralgia	N	0 (0)	1 (1.3)
Eye allergy	Y	0 (0)	1 (1.3)
Asthenopia	N	0 (0)	1 (1.3)
Blepharitis	Y	0 (0)	1 (1.3)
Nasal congestion	Y	0 (0)	1 (1.3)
Neck pain	N	0 (0)	1 (1.3)
Musculoskeletal pain	N	0 (0)	1 (1.3)
Administration site oedema	Y	0 (0)	1 (1.3)
Eye discharge	N	0 (0)	1 (1.3)
Photophobia	Y	0 (0)	1 (1.3)
Rash maculo-papular	N	0 (0)	1 (1.3)
Periorbital swelling	N	0 (0)	1 (1.3)
Hyperhidrosis	N	0 (0)	1 (1.3)
Nephrostomy	N	0 (0)	1 (1.3)
Nasal obstruction	N	0 (0)	1 (1.3)
Pruritus allergic	Y	0 (0)	1 (1.3)
Rhinorrhoea	Y	0 (0)	1 (1.3)
Nasopharyngitis	N	2 (2.1)	0 (0)

AS, Active Surveillance. SN, Spontaneous Notification. * Labeled according to SmPC/monograph.

**Table 5 healthcare-10-02182-t005:** Most frequently reported ADRs in artificial tears databases.

Artificial Tears			
PT	Labeled *	AS, n (%)	SN, n (%)
Burning sensation	Y	26 (74.3)	11 (42.3)
Vision blurred	Y	7 (20)	2 (7.7)
Ocular hyperaemia	N	0 (0)	2 (7.7)
Lacrimation increased	N	0 (0)	2 (7.7)
Ageusia	N	0 (0)	1 (3.8)
Visual impairment	N	0 (0)	1 (3.8)
Anosmia	N	0 (0)	1 (3.8)
Dry eye	N	0 (0)	1 (3.8)
Eye pruritus	N	0 (0)	1 (3.8)
Instillation site discharge	N	0 (0)	1 (3.8)
Eye discharge	N	0 (0)	1 (3.8)
Abnormal sensation in eye	N	0 (0)	1 (3.8)
Foreign body sensation in eyes	N	0 (0)	1 (3.8)
Ocular pain	N	1 (2.9)	0 (0)
Headache	N	1 (2.9)	0 (0)

AS, Active Surveillance. SN, Spontaneous Notification. * Labeled according to SmPC/monograph.

## Data Availability

The data underlying this article will be shared on reasonable request to the corresponding author.

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
