# Peer review of "Underreporting and Triggering Factors for Reporting ADRs of Two Ophthalmic Drugs: A Comparison between Spontaneous Reports and Active Pharmacovigilance Databases"

_healthcare, 2022, doi:10.3390/healthcare10112182_

Round 1

Reviewer 1 Report

This article indicates the rate of underreporting of adverse drug reactions (2 types of ophthalmological drugs) in spontaneous reports. I consider that the work is especially relevant to have an overview of both pharmacovigilance and spontaneous reports of adverse drug reactions, carried out by ophthalmologists, in Latin countries.

Author Response

Thanks for the feedback

Reviewer 2 Report

I would like to congratulate the Authors for their interesting work.

English level and style are satisfactory and the article has an overall good readability.

The introduction provides sufficient background and a rationale for this study.

Methods: are data of spontaneous reporting publicly available? Could you specify the repository in which they are consultable?

line 113: specify SmPC

could you specify inclusion and exclusion criteria for subjects enrolled in active surveillance?

Table 3: Please include the P values for each line, for antiglaucoma and artificial tears.

the type of antiglaucoma drugs and artificial tears are not specified. Different medications with different active ingredients may cause ADRs with different frequencies. 

Subjects enrolled in the active surveillance program could have used by chance products related to a higher rate of ADRs than those used by subjects of the spontaneous reporting. How could you manage this important bias?

How did the Authors choose the sample size for active surveillance?

Although the relevance of the reported subject, some issues need to be addressed. 

Author Response

Methods: are data of spontaneous reporting publicly available? Could you specify the repository in which they are consultable?

They are not public; however, they may be shared with a request and prior authorization from Laboratorios Sophia S.A. of C.V.

line 113: specify SmPC

in line 113 it appears,” The underreporting ratio was calculated using the formula mentioned below”, however, for the underreporting ratio the SmPC is not necessary.

could you specify inclusion and exclusion criteria for subjects enrolled in active surveillance?

Done, added in lines 84 to 85

Table 3: Please include the P values for each line, for antiglaucoma and artificial tears.

Statistical tests were not carried out on all the lines in Table 3, and some of them were on more than one line, for example, severity (3 lines were considered: mild, moderate, and severe), the statistics carried out from that table is mentioned in lines 163 to 169

the type of antiglaucoma drugs and artificial tears are not specified. Different medications with different active ingredients may cause ADRs with different frequencies. 

Done, added in lines 14 to 15, and 64 to 65

Subjects enrolled in the active surveillance program could have used by chance products related to a higher rate of ADRs than those used by subjects of the spontaneous reporting. How could you manage this important bias?

It was not part of the scope of the article to evaluate products that could trigger an increase in the rate of ADRs; however, active surveillance and spontaneous reporting patients have the same chance of using drugs that increase ADRs.

How did the Authors choose the sample size for active surveillance?

There was no defined sample size; it was through the patients collected during a year.

Reviewer 3 Report

Overall: the biggest issue with this project is the design, why are you only including antiglaucoma and artificial tears when there are so many ophthalmic drug classes? This is why you detected a very low number of events and perhaps makes your results difficult to interpret. Other comments related to style, writing, and following journal style are below. You need more work to better represent your results. 

1-    Design: why do you only analyze artificial tears vs antiglaucoma meds? These are not equivalent as antiglaucoma is therapeutic and artificial tears can be cosmetic in many instances. Another thing, what about ophthalmic antibiotics? Steroids? These are very common classes and are used frequently. Maybe consider the title for antiglaucoma and artificial tears.

2-    Writing: inconsistent writing of numbers, you need to add commas for everything third digit. E.g. 1,000,000, not consistent in all manuscript 

3-    Design: there are a varying patterns of practice of medicine and reporting ADRs in the countries included, so the results can’t be generalized.

4-    Revise all the references in the texts and at the end of the manuscript based on the MDPI Healthcare style. The reference should be in a bracket at the end of the sentence before the period. Make sure to adhere to the author’s guideline 

5-     The collection of 31 ADRs begins from the early stages of drug development, such as during randomized clin-32 ical trials (RCT) and although they are considered the cornerstone of drug safety and ef-33 efficacy evaluation, revise to “ ADRs are identified during early preclinical stages and during the randomized controlled trials. However, it is challenging to detect rare ADRs, thus, post-marketing studies are essential.

6-    Delete this part “ these trials include controlled and monitored populations for short pe-34 riods2,3” 

7-    I could not understand this sentence, it is confusing “It is characterized by no active measures to scrutinize ADRs apart from the encour-40 agement of health professionals and patients to report them; therefore, such reports de-41 pend entirely on their initiative and motivation7.

8-    The results are difficult to interpret, you should consider simplifying what that means. Your audience is clinicians. I highly recommend that you add all the classes of ophthalmic medications and perform subgroups based on each therapeutic class. 

9-    You need to re-organize your discussion based on this article structure. Hess DR. How to write an effective discussion. Respir Care. 2004 Oct;49(10):1238-41. PMID: 15447810.

Author Response

-    Design: why do you only analyze artificial tears vs antiglaucoma meds? These are not equivalent as antiglaucoma is therapeutic and artificial tears can be cosmetic in many instances. Another thing, what about ophthalmic antibiotics? Steroids? These are very common classes and are used frequently. Maybe consider the title for antiglaucoma and artificial tears.

The study aimed to calculate the underreporting ratio for two different medications and characterize the features influencing the reporting of adverse drug reactions (ADRs) in spontaneous reporting, regardless of the type of drug compared; In addition, using two drugs with different safety profiles was useful since almost twice as many antiglaucoma ADRs were found compared to artificial tear because the severity/seriousness of adverse reactions in the artificial tear database is lower than for antiglaucoma. Lines 234 to 236.

At this time, other active pharmacovigilance studies are being carried out to be able to evaluate other drugs; however, at the moment, we do not have active pharmacovigilance studies of these drugs to be able to make the comparison.

2-    Writing: inconsistent writing of numbers, you need to add commas for everything third digit. E.g. 1,000,000, not consistent in all manuscript 

Done

3-    Design: there are a varying patterns of practice of medicine and reporting ADRs in the countries included, so the results can’t be generalized.

The wording was changed to avoid generalizing that the pharmacovigilance system is a possible cause of the phenomena of non-reporting since there is not enough information to support it.

The change made is consistent with the study objective and shows relevant information regarding the concern of a low underreporting rate.

Added in lines 218 to 224

4-    Revise all the references in the texts and at the end of the manuscript based on the MDPI Healthcare style. The reference should be in a bracket at the end of the sentence before the period. Make sure to adhere to the author’s guideline 

Done

5-     The collection of ADRs begins from the early stages of drug development, such as during randomized clinical trials (RCT) and although they are considered the cornerstone of drug safety and efficacy evaluation, revise to “ ADRs are identified during early preclinical stages and during the randomized controlled trials. However, it is challenging to detect rare ADRs, thus, post-marketing studies are essential.

We decided not to change this sentence, due it contains relevant information.

6-    Delete this part “ these trials include controlled and monitored populations for short periods” 

We decided not to eliminate this sentence, due it contains relevant information. Since clinical studies are usually relatively short and patients have various inclusion and exclusion criteria that do not allow sufficient evidence to be collected, unlike post-marketing pharmacovigilance.

7-    I could not understand this sentence, it is confusing “It is characterized by no active measures to scrutinize ADRs apart from the encouragement of health professionals and patients to report them; therefore, such reports depend entirely on their initiative and motivation.

There is no action than the own decision of health professionals and patients to notify ADRs

The wording was modified to clarify this sentence.   Lines 43 to 45

8-    The results are difficult to interpret, you should consider simplifying what that means. Your audience is clinicians. I highly recommend that you add all the classes of ophthalmic medications and perform subgroups based on each therapeutic class. 

In this study were compared a fixed combination of 0.5% timolol + 0.2% brimonidine + 2.0% dorzolamide (antiglaucoma) and a fixed combination of sodium hyaluronate 0.1% + chondroitin sulfate 0.18% (artificial tear) through two pharmacovigilance methods, active surveillance and spontaneous reporting.

There are no therapeutic groups, only two drugs, a fixed combination of 0.5% timolol + 0.2% brimonidine + 2.0% dorzolamide (antiglaucoma) and a fixed combination of sodium hyaluronate 0.1% + chondroitin sulfate 0.18% (artificial tear)

Round 2

Reviewer 3 Report

Dear authors, thank you for your efforts to improve the paper, yet there are major limitations.

comments are made in the first round of revision to improve the paper overall and revise the title of the manuscript to reflect the data used in the study of glaucoma and artificial tears (now I feel the title is very broad and does not reflect all ophthalmic medications which can be misleading). 

There were sentences I suggested removing and/or revising, however, the reviewer's suggestions to improve the paper were dismissed. 

I don't have further comments

best 

Author Response

We appreciate the thoughtful and thorough provided by the reviewers to improve this work. In addition, we are thankful for their time and effort in providing quality peer reviews. 

Comments are made in the first round of revision to improve the paper overall and revise the title of the manuscript to reflect the data used in the study of glaucoma and artificial tears (now I feel the title is very broad and does not reflect all ophthalmic medications which can be misleading). 

The title was changed because the previous one gives the appearance that a large number of ophthalmic medications were analyzed.

“Underreporting and triggering factors for two ophthalmic drugs’ ADR notification; a comparison between spontaneous reporting and active pharmacovigilance databases.”

There were sentences I suggested removing and/or revising, however, the reviewer's suggestions to improve the paper were dismissed. 

We have reconsidered the recommendations and made the following changes:

Lines 33 to 36 were changed to “ADRs are identified during early preclinical stages and randomized controlled trials; however, it is challenging to detect rare ADRs, thus, post-marketing studies are crucial”

And the part “these trials include controlled and monitored populations for short periods” was deleted